# Capillary Force-Driven Quantitative Plasma Separation Method for Application of Whole Blood Detection Microfluidic Chip

**DOI:** 10.3390/mi15050619

**Published:** 2024-05-01

**Authors:** Xiaohua Fang, Cuimin Sun, Peng Dai, Zhaokun Xian, Wenyun Su, Chaowen Zheng, Dong Xing, Xiaotian Xu, Hui You

**Affiliations:** School of Mechanical Engineering, Guangxi University, Nanning 530004, China; 2111301016@st.gxu.edu.cn (X.F.); cmsun@gxu.edu.cn (C.S.); pengdai@st.gxu.edu.cn (P.D.); zhaokunxian@st.gxu.edu.cn (Z.X.); swy8138@st.gxu.edu.cn (W.S.); cwzheng@st.gxu.edu.cn (C.Z.); xdd0721@st.gxu.edu.cn (D.X.)

**Keywords:** whole-blood, quantitative separation, filter, microfluidic chip

## Abstract

Separating plasma or serum from blood is essential for precise testing. However, extracting precise plasma quantities outside the laboratory poses challenges. A recent study has introduced a capillary force-driven membrane filtration technique to accurately separate small plasma volumes. This method efficiently isolates 100–200 μL of pure human whole blood with a 48% hematocrit, resulting in 5–30 μL of plasma with less than a 10% margin of error. The entire process is completed within 20 min, offering a simple and cost-effective approach to blood separation. This study has successfully addressed the bottleneck in self-service POCT, ensuring testing accuracy. This innovative method shows promise for clinical diagnostics and point-of-care testing.

## 1. Introduction

Whole blood testing is essential for detecting and diagnosing various diseases, as blood plasma, which contains nucleic acids, proteins, and metabolites, serves as the primary carrier for disease biomarkers [1,2]. However, the presence of blood cells and substances released during hemolysis can impact the accuracy of biomarker detection, making plasma separation a necessary preprocessing step [3]. The removal of blood cells is crucial for improving testing accuracy, while quantitative sampling techniques ensure reliability and repeatability [4,5]. Despite this, current whole blood sampling techniques lack quantification, which hinders the development of self-service point-of-care testing.

Recent research has focused on micro-volume plasma separation techniques, which can be integrated into microfluidic devices to simplify sampling procedures and reduce the size of analytical instruments. This advancement opens up new possibilities for integrating whole blood testing into point-of-care testing applications [6,7,8,9,10].

Plasma separation methods are categorized into active and passive methods [9]. Active methods employ centrifugation [11,12], acoustic waves [13,14], electrophoresis [15,16], or magnetic [17] forces for rapid and pure plasma separation but require external power sources or specific substances [18]. In contrast, passive methods separate plasma using fluid dynamics or microstructures without the need for external power, enabling integration with portable microfluidic devices and reducing costs [19,20,21,22,23,24].

Passive plasma separation methods, such as membrane-based filtration, are valued for their simplicity, ease of manufacturing, scalability, and portability, making them suitable for integration with POCT microfluidic devices [25,26]. However, membrane-based filtration methods commonly encounter challenges. These challenges include red blood cell hematocrit leading to plasma contamination, blood cell clogging membranes, and red blood cell leakage reducing separation purity. To tackle these issues, various methods have been proposed by research teams.For instance, Liu et al. employed an asymmetric porous polysulfone membrane to filter virus-containing whole blood. This approach effectively reduces red blood cell contamination of plasma. However, it is only suitable for small-scale whole blood filtration with low plasma collection rates [27]. Baillargeon et al. introduced a pre-filter before the membrane, using a polyester material with white blood cell adsorption capabilities to enhance filtration efficiency. This method also reduces hemolysis caused by hematocrit but results in additional plasma loss and lowers the final plasma collection rate [21]. Gao et al. utilized narrow capillaries to enhance plasma separation efficiency, reducing residual plasma volume, and increasing collection rates [28]. Nevertheless, previous studies were confined to laboratory settings, necessitating manual tools for precise plasma separation [6,7,8,29,30,31]. In point-of-care testing (POCT), where fingertip blood sampling is prevalent, the lack of specialized equipment impacts result accuracy. Current methods that rely on professional equipment deviate from the original purpose of POCT [32]. Therefore, there is an urgent need for a simple, fast, and cost-effective quantitative plasma separation method integrated into microfluidic chips to enable practical POCT use.

This study introduces a new, straightforward, and speedy approach for quantitatively extracting plasma from a small quantity of undiluted whole blood, addressing a challenge in self-contained chip technology. The membrane filtration process is hindered by the deformation and buildup of red blood cells, which gradually block the filter pores and impede filtration. Consequently, filtration slows down when a certain volume is reached, ensuring a consistent volume of plasma is obtained regardless of the input amount of whole blood. This improves the accuracy of subsequent biomarker detection. Unlike traditional plasma separation methods, this approach does not rely on external power sources or specialized personnel for operation, allowing for the separation of a specified amount of whole blood in just 20 min. Nevertheless, further research is necessary to assess its suitability for various environmental conditions.

## 2. Experimental Fracture

### 2.1. Quantitative Principle

Current plasma separation methods utilizing membrane filtration primarily aim to improve separation efficiency. While many researchers focus on speed, few studies highlight the importance of precise quantitative plasma acquisition to enhance detection accuracy. A prevalent challenge in membrane filtration techniques is the gradual pore blockage, which hampers plasma flow, leading to incomplete filtration and excessive plasma residue.

This phenomenon can be elucidated based on the filtration cake filtration principle. By employing the filtration rate balance equation, we can derive the filtration rate [33]:(1)$$\frac{dV}{d\tau }=\frac{A^{2}\Delta P}{r\mu v(V+V_{e})}$$

The equation involves *V* for filtered plasma volume, τ for filtration time, *A* for effective filtration area, $V_{e}$ for equivalent filtrate volume achieved when filtration membrane resistance equals filter cake resistance, *r* for filter cake specific resistance influenced by material and compressibility. In plasma separation research, the filter cake is primarily red blood cells, μ is plasma viscosity, *v* is volume of red blood cell filter cake per unit plasma volume, and ΔP is filtration process driving force, particularly capillary force in microchannels in this study. According to the Young-Laplace equation, capillary force in rectangular cross-section microchannels is determined [34]:(2)$$\Delta P=2\gamma \cos\theta (\frac{1}{h}+\frac{1}{w})$$

In the equation, γ represents the surface tension of the liquid at the concave meniscus in the microchannel (the gas-liquid interface shown in Figure 1c). θ denotes the contact angle of the microchannel surface, while *h* and *w* stand for the height and width of the microchannel, respectively. According to the formula, when the liquid and the contact angle remain constant, the capillary force’s magnitude is solely dependent on the microchannel dimensions and will increase as these dimensions decrease. In this study, depicted in Figure 1, the microchannel comprises grooves machined between the top cover plate and the substrate. It has a height of 80 μm and a width that can be approximated as the diameter of the machined circular groove, approximately 38 mm. Consequently, we can assume that the width of the microchannel is significantly greater than its height. Therefore, the capillary force within the microchannel can be expressed as follows [9]:(3)$$\Delta P=\frac{2\gamma \cos\theta }{h}\;$$

To determine the plasma volume derived from the separation process, we integrate Formula (1) following a conversion on both sides. This yields the formula for plasma volume *V* in relation to filtration time τ. By substituting the outcome of Formula (3) into this equation, we establish the correlation between plasma volume *V* and filtration time τ.
(4)$$V=\sqrt{\frac{2A^{2}\tau \gamma \cos\theta }{r\mu vh}+V_{e}^{2}}-V_{e}$$

When the effective filtration area *A* remains constant, the plasma collection volume *V* is approximately proportional to the square root of the filtration time τ. This correlation mirrors the connection observed in the membrane filtration experiments conducted by Gao et al., where the volume of plasma collected was linked to the duration of the process [28].

Based on Formula (1), as the plasma collection volume *V* increases during the plasma separation process, the filtration rate gradually decreases. Once the rate of change in plasma volume per unit time *q* is 5% or less, the separation process is considered to have reached a stable state. This moment is denoted as $\tau_{q}$, with the resulting plasma volume labeled as $V_{q}$.
(5)$$q=\frac{\Delta V}{V}=\frac{A^{2}\Delta P}{r\mu vV(V+V_{e})}⩽5\%$$

The steady-state relationship between plasma collection volume $V_{q}$ and effective filtration area *A* can be defined as:(6)$$V_{q}=\sqrt{10kA^{2}+0.25V_{e}^{2}}-0.5V_{e}$$

In this study, the filtering constant k=2ΔP/rμv is defined. The determined value of *k* indicates the rate at which the plasma collection volume $V_{q}$ changes with respect to the effective filtration area *A*. It is observed that this rate reaches a steady state during separation.
(7)$$\frac{dV_{q}}{dA}=\frac{10kA}{\sqrt{10KA^{2}+0.25V_{e}^{2}}}>0$$

The second derivative of the plasma collection volume $V_{q}$ with respect to the effective filtration area *A* reaches a steady state during separation:(8)$$\frac{d^{2}V_{q}}{dA^{2}}=\frac{2.5KV_{e}^{2}}{\sqrt{{\left(10KA^{2}+0.25V_{e}^{2}\right)}^{3}}}>0$$

The calculation above shows that the plasma collection volume $V_{q}$ stabilizes when plotted against the effective filtration area *A*. The relationship is concave, indicating that *V* qreaches a stable state as *A* continuously increases.

### 2.2. Device Design

The plasma separation device developed in this study operates on the principle of membrane filtration. In this process, blood cells gradually obstruct the pores of the filter, impeding further filtration. Illustrated in Figure 1, the device is uncomplicated in structure, conducive to mass production, and primarily comprises a bottom plate, a top cover plate, a filter membrane, and a sealing ring. The circular filter membrane is affixed to the top cover plate via a circular sealing ring. The bottom plate features a 37 mm diameter circular groove, 80 μm deep, with a central protrusion measuring 0.58 mm in height and 6 mm in diameter. This protrusion includes four notches at its periphery to ensure smooth plasma flow. Upon bonding the top cover plate and bottom plate, a liquid storage groove of approximately 80 μm is created in the middle layer to store plasma and leverage capillary force as a filtration driving mechanism.

To seal the top cover plate and bottom plate, BSA (bovine serum albumin) is employed to reduce the contact angle between the microchannel and plasma, enhance its hydrophilicity, and facilitate seamless plasma flow within the microchannel. Furthermore, there are four pressure balance holes at the four corners of the upper cover plate, which are connected to the microchannel to balance the microchannel with external air pressure and ensure the smooth flow of plasma.

This study employs an asymmetric polyethersulfone (PES) membrane characterized by an uneven pore structure, where the pores in the upper layer are larger than those in the lower layer. During filtration, blood cells access the larger upper layer pores but are unable to traverse the smaller lower layer pores, effectively minimizing blood cell rupture while being susceptible to red blood cell blockages. To address red blood cell seepage, an interference fit sealing technique is utilized, as depicted in Figure 1. The filter membrane is securely held in place through the clamping force between the base of the mounting hole on the top cover plate and the sealing ring, with the sealing ring slightly larger than the mounting hole to ensure stable installation via an interference fit. By compressing the edges of the filter membrane, the ingress of blood cells into the liquid storage groove along the membrane’s edge is prevented.

This study uses blood cell blockage in plasma filtration experiments to achieve precise filtration. By modifying the shape and area of the filtration chamber, we determine the correlation between the plasma volume obtained and the chamber area. This approach enables the development of a device that accurately acquires the desired plasma volume, enhancing the detection of biomarkers.

### 2.3. Device Fabrication

The upper and lower substrates of this device are constructed from colorless and transparent polymethyl methacrylate (PMMA) known for its high light transmittance and excellent biocompatibility. PMMA is chosen for its ease of processing and manufacturing, which aids in observing plasma separation processes and estimating plasma volume accurately.

The device utilizes an asymmetric polysulfone membrane (PSM0180-B, Cobetter, Hangzhou, China) with a thickness of 350 μm. This membrane features varying pore sizes on its upper and lower surfaces −4.5 μm on the upper surface and 1.8 μm on the lower surface. For bonding, a 60 μm thick double-sided tape (SD-D02, Wenhao, Suzhou, China) with outstanding biocompatibility is employed. This tape can directly interact with biological materials without any biotoxicity concerns.

The assembly diagram of the equipment is depicted in the figure, with the main structural dimensions specified in Figure 2.

To begin, employ a laser cutting machine (HZZ-V3000S, HZZ, Guangzhou, China) to cut PMMA sheets measuring 3 mm and 5 mm in thickness into 40 × 40 mm^2^ substrates. For the 3 mm thick substrate, utilize a precision CNC milling (SYIL X5, Sharpe CNC, Ningbo, China) machine to fashion a groove and a central protrusion. The groove should have a depth of 80 μm, while the protrusion should possess a diameter of 6 mm and a height of 580 μm, resulting in a height disparity of 500 μm between the protrusion and the edge. Similarly, for the 5 mm thick cover plate, employ a precision CNC milling machine to craft a 10 mm diameter groove for installing a filter membrane, with a processing depth of 4.5 mm. Afterward, four pressure balance holes are created on the top cover plate using a laser, and three filter holes are cut out on the filter membrane installation groove, providing space for the installation of the filter membrane and sealing ring. Following this step, any remaining debris and cutting fluid on the upper and lower substrates are eliminated through ultrasonic cleaning (F-100SD, Fuyang, Shandong, China).

Subsequent to cleaning, the grooves are immersed in a 3% bovine serum albumin (BSA) solution and then incubated at 37 °C for 2 h in an oven. This process effectively enhances the hydrophilicity of the groove surface, maintaining it at a heightened level for an extended duration. Post-sealing treatment, the contact angle on the groove surface transitions from 77.8 degrees to 27 degrees, persisting in this highly hydrophilic state for up to a week without reverting to a low hydrophilicity state.

Following this, a 60 μm double-sided adhesive tape (SD-D02) is cut to match the shape of the top cover plate, retaining the filter and pressure balance holes, and affixed to the underside of the top cover plate. Subsequently, the filter membrane is cut into a 10 mm diameter circle using surgical scissors. A sealing ring (PMMA), crafted with precision using a CNC machine and laser, is employed to secure the filter membrane in the mounting hole. The sealing ring possesses an outer diameter of 10.32 mm and a 6 mm diameter filter hole corresponding to the protrusion on the middle and lower substrates, resulting in an effective filtration area of $\pi *3^{2}$ mm^2^.

To finalize the assembly, the top cover plate and bottom plate are bonded together using double-sided tape. The overall dimensions of the device measure 40 × 40 × 8 mm^3^.

The equipment produced through the aforementioned steps is illustrated in Figure 2. The materials utilized consist of PMMA, priced at roughly 0.041 dollar, asymmetric polysulfone film, priced at approximately 0.0096 dollar, 60 μm double-sided tape, costing about 0.21 dollar. The total production cost for the entire device is approximately 0.26 dollar. In general, the manufacturing process for this device is straightforward, the raw materials are readily available, and the costs are relatively low, providing an advantage for its potential commercial applications.

### 2.4. Blood Sample Preparation

We utilized human whole blood to evaluate the device’s performance. Blood was obtained via venous blood sampling from a healthy adult male volunteer aged 25. The whole blood was anticoagulated with ethylenediaminetetraacetic acid dipotassium salt (EDTA-K2) immediately after extraction. This process involved using a commercially available 10mL vacuum blood collection tube, and the samples were then stored at 4 °C.

### 2.5. Experiment Protocols

Before commencing the experiment, we initially warm the stored whole blood in a 37 °C water bath to prevent platelet cold agglutination at lower temperatures. This precaution is necessary as platelet cold agglutination could potentially disrupt the filtration process and bring the sample closer to real-time detection. Using a pipette (10–100 μL, Thermo Fisher Scientific, Waltham, MA, USA), we carefully transfer a specific volume of whole blood from the collection tube to the separation device. All experiments are carried out at room temperature (25 ± 2 °C) under consistent conditions.

The separation of plasma is visually monitored using a camera, and we deem the process complete when the movement of the plasma meniscus in the reservoir ceases. Images are captured at 30-s intervals from the recorded video. Utilizing ImageJ 1.53e software, we compute the area (*S*) occupied by the plasma in the reservoir. Given that the height of the microchannel remains constant (*h*), the calculated plasma volume (*V*) is determined as the product of the area and the height (V=S∗h).

Multiple comparative experiments are conducted employing various effective filtration areas while maintaining other parameters constant. By documenting the final plasma volumes acquired with different filtration areas, we establish a correlation between the collected plasma volume and the effective filtration area based on the experimental data. This analysis aids in determining the optimal effective filtration area required to achieve the desired volume of plasma.

## 3. Results and Discussionl Fracture

### 3.1. Plasma Separation Quality

Red blood cells constitute the largest and most significant component of whole blood (with a hematocrit range of 40% to 50% for healthy adult males and 36% to 45% for females). As a result, the residual blood cell content serves as a critical indicator for evaluating the effectiveness of plasma separation.

During the plasma separation process, the effectiveness of edge leakage prevention by the sealing ring is evident through the transparent and observable sealing ring, ensuring that no red blood cells leak into the submembrane layer. The separated plasma, colorless and transparent, indicates minimal to no blood cell leakage into the microchannels. To assess the separation quality of our new method, we conduct blood cell counting on the separated plasma using a red blood cell counting plate, comparing it with plasma obtained through centrifugation.

Blood cell counting on the extracted plasma involves a five-point sampling method, with five random samples taken from each of the three parallel experiments. Each sample point, covering an area of approximately 250 μm^2^ with a microchannel height of around 80 μm, results in an estimated plasma volume of 0.02 μL within the sample point area. This allows us to determine the blood cell residue rate in the extracted plasma, which is found to be 0.202%, 0.266%, and 0.248% for the three parallel experiments, respectively.

In conclusion, the extracted plasma contains 0.2% red blood cells, indicating a red blood cell capture efficiency of 99.8% for the new device. In contrast, the plasma obtained through centrifugation shows no red blood cell content, highlighting the higher red blood cell capture efficiency of the new method. The quality of the extracted plasma closely matches that of the centrifugation-obtained plasma, demonstrating excellent separation quality.

The process of separation is depicted in Figure 3. It is evident from the illustration that there is no red leakage from the side sealed with an O-ring. The filtered plasma is transparent and devoid of blood stains. Furthermore, the two filtration devices depicted in the figure share the same effective filtration area. Despite varying volumes of whole blood being introduced to the inlet of the devices during the filtration experiment, the volumes of plasma obtained remained consistent. This observation is drawn from the fact that, although the residual volumes of whole blood in the inlets differed, the areas occupied by plasma in the microchannels were approximately equal.

The quality of separation is also linked to the extraction of free hemoglobin in the plasma. An elevation in dissociative hemoglobin levels may be indicative of hemolysis. Experimental observations of the extracted plasma visually show a color that is not pinkish but instead colorless and transparent. This suggests that plasma separation in the device does not induce hemolysis, or does so only to a minimal extent.

### 3.2. Relationship between Plasma Volume and Effective Filtration Area

To explore the relationship between the volume of plasma collected and the effective filtration area, we conducted filtration experiments using devices featuring varying effective filtration areas. Our findings confirmed that the volume of plasma obtained differs when an excessive amount of whole blood is introduced, depending on the effective filtration area of the membranes. Figure 4 illustrates the curves depicting changes in plasma collection volume over time for different effective filtration area sizes.

It is crucial to highlight that when the effective filtration area is below 7 mm^2^, capillary forces exerted on the inlet tube wall impede the filtration process. This obstruction hampers the generation of a sufficient driving force for further filtration, even post hydrophilic treatment of the microchannels.

Multiple experiments employing different effective filtration areas consistently reveal a common trend: the filtration rate gradually decreases as the process progresses, eventually approaching zero. Towards the end of the filtration process, some whole blood remains at the inlet, indicating that as filtration continues, blood cells increasingly block the filter pores, causing a decline in the filtration rate. Despite the presence of unfiltered whole blood in the filter chamber, the collected plasma volume gradually stabilizes, consistent with the observations from Formula (4). The relationship between plasma volume and time approximates a square root function.

This study introduces a quantitative filtration method that analyzes the relationship between final plasma volume and effective filtration area. By controlling the size of the filtration area, quantitative plasma filtration is achieved. Estimations for the required effective filtration area to obtain specific plasma volumes can be derived from Figure 5. To enable precise plasma filtration, a filtration device has been developed.

Based on Figure 5, the relationship between the stable collection volume of separated plasma and the effective filtration area when separating whole blood using membrane filtration principles roughly follows the trend shown in the figure. These results are consistent with earlier calculations. The dashed line in the figure represents the theoretical trend line obtained through dimensionless processing using Formula (6), which aligns with the experimental data trend. Therefore, based on the aforementioned trend curve, we can estimate the size of the effective filtration area corresponding to the volume of plasma obtained quantitatively. Furthermore, these studies were conducted under the same external environmental conditions. Therefore, further research is needed to explore applications in various environments, especially under extreme conditions.

In conclusion, we have validated the relationship between the volume of plasma collected and the time in the capillary force-driven plasma separation device. Additionally, we have confirmed the relationship between the steady-state collection of plasma and the effective filtration area, aligning with the earlier theoretical derivation. Based on this relationship, we propose an application: controlling the steady-state collection of plasma by adjusting the size of the effective filtration area. This approach aims to achieve quantitative plasma separation without the need for external driving forces.

### 3.3. Quantitative Filtration Experiment

Based on the experimental results presented in Figure 5, we can determine the approximate range of effective filtration area necessary to accurately achieve a specific volume of plasma and tailor the size of the filtration device accordingly. The experimental blood plasma filtration was carried out using volumes ranging from 5 to 30 μL, with various gradients of effective filtration areas being tested. The selected effective filtration areas were 15 mm^2^, 22.5 mm^2^, 25 mm^2^, 29 mm^2^, and 32.5 mm^2^, corresponding to plasma collection volumes of 5 μL, 10 μL, 15 μL, 20 μL, and 30 μL, respectively. Each device was loaded with an excess of 100 to 200 μL of whole blood. The results of these experiments are depicted in Figure 6.

From Figure 6, it is evident that the quantity of plasma obtained from a 5 to 30 microliter quantitative plasma filtration device closely aligns with the target amount, with a small margin of error within ±10%, falling within an acceptable range of error. Experimental evidence indicates that by adjusting the size of the effective filtration area according to the stable collection volume of plasma, it is feasible to attain the desired plasma volume, albeit with slight discrepancies. This method is of practical significance for microfluidic chips and plays a role in addressing the challenge of quantitative sample injection in self-testing whole blood chips.

## 4. Conclusions

Based on the cake layer filtration mechanism, we have developed a quantitative plasma separation method driven by capillary force and membrane filtration. This method allows for the precise separation of microliter-level plasma even in non-laboratory conditions, showing promise for applications in microfluidics. The device designed in this study boasts a simple structure, low cost, and the potential for mass production.

This device can achieve quantitative plasma separation within the range of 5 to 30 μL, completing the process in 20 min with a quantitative error of less than 10% and a red blood cell capture efficiency of 99.8%. In contrast to prior studies, this research has successfully achieved quantitative plasma separation in non-laboratory settings, obviating the requirement for supplementary equipment and relying solely on the device’s intrinsic characteristics to achieve the desired plasma separation outcome.

In essence, this plasma separation device facilitates accurate plasma separation, thereby improving detection precision. It is characterized by its uncomplicated structure, affordability, absence of intricate microstructures, and ease of integration into microfluidic devices. These features lay the groundwork for the advancement of self-service Point-of-Care Testing (POCT) and hold significant commercial potential in clinical medicine and POC testing.

## Figures and Tables

**Figure 1 micromachines-15-00619-f001:**
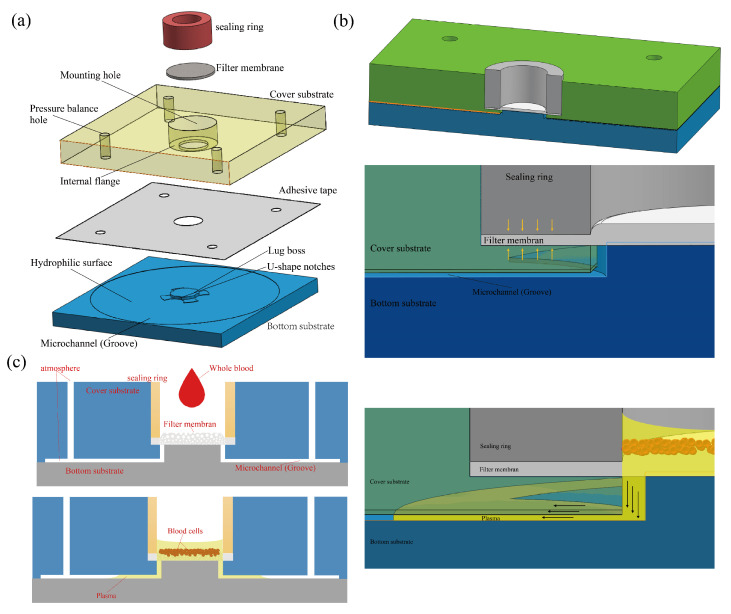
The schematic illustrates the device design and filtration principle (**a**) the device structure and assembly diagram (**b**) the cross-section and local schematic of the filtration slot (**c**) schematic of the filtration process with the plasma filling the reservoir slot, and indicate that the pressure balance hole connects the microchannel with the external balance of internal and external pressure.

**Figure 2 micromachines-15-00619-f002:**
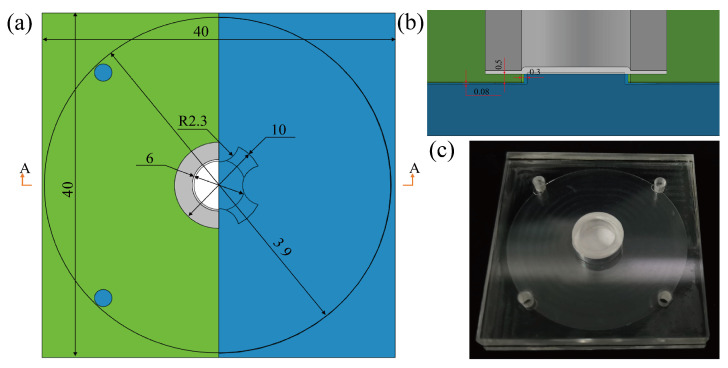
The structural diagram and dimension annotations (unit: mm) of the designed device, as well as the physical image of the device: (**a**) top view of the device to show the internal structure, depicted as a half-section diagram, (**b**) sectional view along A-A showing dimension annotations of microchannels, flanges, and boss gaps, (**c**) physical image of the device.

**Figure 3 micromachines-15-00619-f003:**
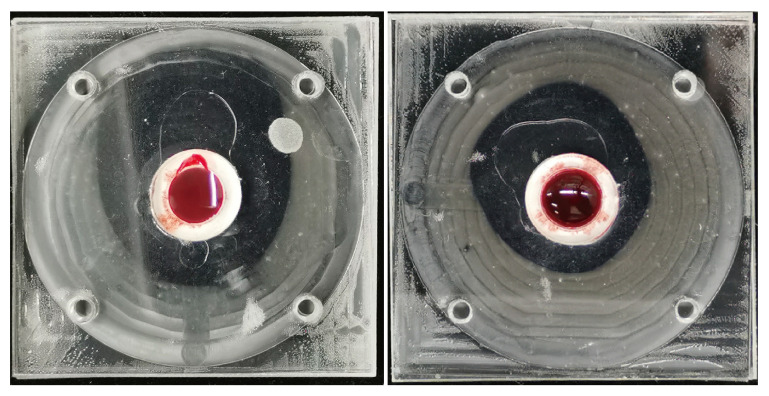
The example of the device during the whole blood separation process shows that there is no blood cell leakage on the side of the filter membrane, there is no plasma contamination, and the plasma volume in the two devices is similar. There is a significant difference in the residual whole blood volume at the injection port.

**Figure 4 micromachines-15-00619-f004:**
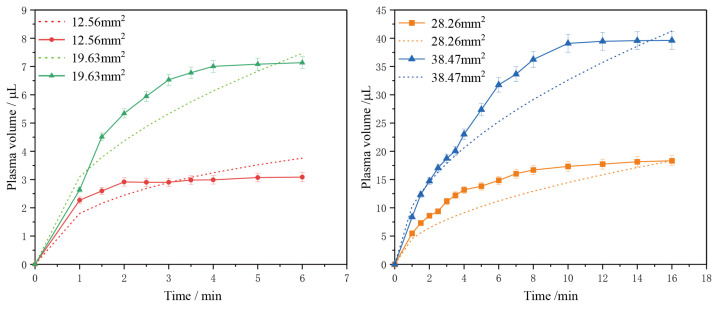
The plasma volume collected by filters with different effective filtration areas shows a trend of change over time. The solid line data represents the average of the results of three parallel experiments with effective filtration areas of 12.56 mm^2^, 19.63 mm^2^, 28.26 mm^2^, and 38.47 mm^2^ in diameter, demonstrating the variation in plasma collection over time. The dashed line represents the theoretical relationship between plasma volume obtained according to the formula and time.

**Figure 5 micromachines-15-00619-f005:**
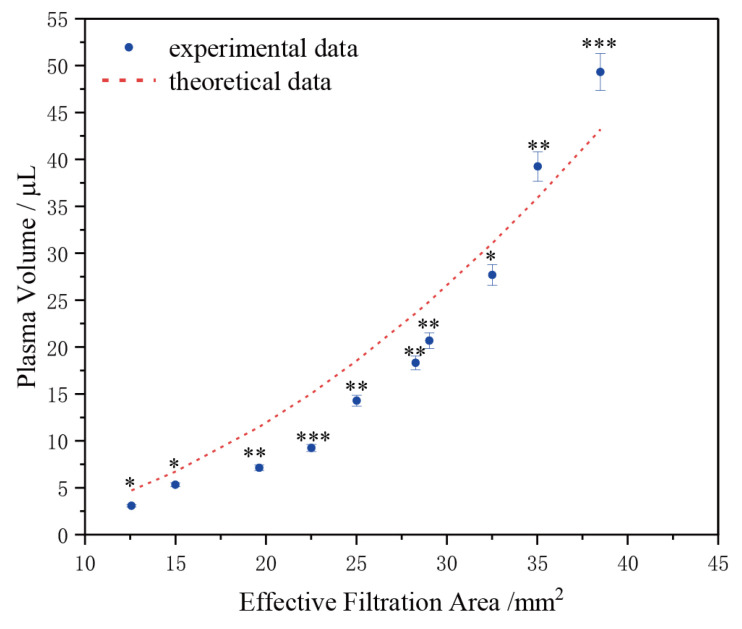
The collected plasma volume stabilizes in relation to the effective filtration area, with the dashed line representing the theoretical predicted curve obtained based on Formula (6). “*” represents the significance level of the data.

**Figure 6 micromachines-15-00619-f006:**
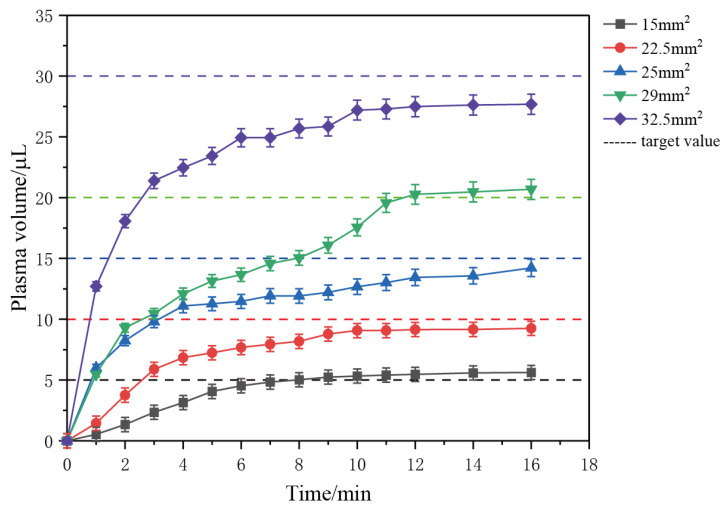
Quantitative experiments collected 5–30 μL of plasma to observe the trend of plasma volume over time, with each data point representing the average of data obtained from two identical devices.

## Data Availability

Data are contained within the article and Appendix A.

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
