# Peer review of "Capillary Force-Driven Quantitative Plasma Separation Method for Application of Whole Blood Detection Microfluidic Chip"

_micromachines, 2024, doi:10.3390/mi15050619_

Round 1

Reviewer 1 Report

Comments and Suggestions for Authors

The authors in this paper present a quantitative method for plasma separation from blood combining membrane filtration with capillary force driving. The microscale methods have potential applications for self-service point-of-care testing outside of traditional laboratory settings.

However, the separation was characterized with the naked eye (“Visual observation indicates that the separated plasma is colorless and transparent, suggesting minimal to no hemolysis occurs during plasma separation in the device.” Lines 198-199). Furthermore, it would be important to make a comparison between the plasma from the reported device with that one obtained by standard centrifugation method.

Another important aspect is checking plasma quality by using biological tests that is crucial for subsequent applications of blood detection microfluidic chip. Unfortunately, this problem is not addressed by the authors.

Additionally, there are a few minor issues in the paper, such as:

Line 83: For clarity, authors should specify which interface they refer to when writing “surface tension”.

Line 88-89: “…the width of the microchannel greatly surpasses its height, …”

The size of the microchannels is not mentioned in the text and it is not clearly read in Figure 1.

Line 91: For easier reading, I suggest explaining how formula 4 was derived.

Line 94-96: “This correlation aligns with previous extensive membrane filtration experiments that have illustrated the relationship between plasma collection volume and time.”

Is "previous" correct? If so, which "previous experiments" do the authors refer to?

Line 125-126: It is not clear from the picture of Figure 1 how the holes are connected to the fluid network and how they balance the pressure.

Line 150: The authors should clarify if 3mm and 5mm refer to the thickness of the PMMA sheets.

Line 173: There is an error in temperature writing “252C”. I think the authors wanted to write "25°C".

Line 175-177: “The separation of plasma is visually monitored using a camera, and we deem the process complete when the movement of the plasma meniscus in the reservoir ceases. Images are captured at 30-second intervals from the recorded video”.

Authors should display videos and/or images of what they claim.

Figure3: It is not clear what the authors want to demonstrate with these images. I suggest explaining it both in the caption and in the text.

Line 201: “…single pores of different diameters…”  It is not clear what this means.

Line 206: “…when the pore diameter is less than 3mm…”

The authors are referring to the area and not the size of the pores, right? If so, they must correct.

Author Response

Thank you very much for taking the time to review this manuscript. Your valuable suggestions are highly constructive and meaningful. Your feedback has been of great help to me, providing significant assistance in both the research and logical aspects of the article. I am very grateful for your input. For detailed information, please refer to the attachment.

Reviewer 2 Report

Comments and Suggestions for Authors

In the manuscript titled "Capillary Force-Driven Quantitative Plasma Separation Method for Application of Whole Blood Detection Microfluidic Chip", the authors present an innovative approach to plasma separation using a microfluidic device driven by capillary forces. This technique promises significant advancements in point-of-care testing (POCT) by enabling rapid, accurate, and cost-effective plasma separation outside traditional laboratory settings. However, the paper requires some enhancements to meet academic standards of micromachines. As reviewer, I offer the following questions:

The manuscript showcases some impressive figures on separation efficiencies and error margins, but it’s a bit weak on the reproducibility front. It’d be really helpful to see more experiments that prove these results aren’t just one-time. Could you try running these tests under different conditions—like with various blood samples or at different room temperatures—to give a fuller picture?

The paper said this technology could be used for clinical pre-testing, which sounds fantastic, but where’s the data to back that up? Including some trial results using real clinical samples would strengthen your argument and help us see how it might work out in the wild.

Also, while it’s great that you highlight the cost-effectiveness and the simplicity of manufacturing this device, the actual numbers are missing. Can you tell the specifics? 

I noticed the paper doesn’t really stack up your new method against the old guards like centrifugation or electrophoresis. Some comparison to show off why your method could be better (or where it might fall short) would strengthen your case.

The device design section could be more specific. In this paper, it fail to describe some crucial details like what materials you’re using for the membrane and the sizes of the pores. Showing these details could give me a better understanding of how your device works.

Finally, your manuscript needs to lay out the statistical analysis methods more clearly. Which tests did you run? What significance levels were considered? 

These revisions are essential for enhancing the manuscript's academic rigor, thereby aligning it more closely with the journal's publication standards.

Comments on the Quality of English Language

Clear to understand, but it could be improved in the introduction and results.

Author Response

(The authors gave the same response as above.)

Round 2

Reviewer 1 Report

Comments and Suggestions for Authors

I thank the authors for all the clarifications.

With the changes made, I believe that the new version of the manuscript can now be published.